# Usefulness of 68-Gallium PET in Type I Gastric Neuroendocrine Neoplasia: A Case Series

**DOI:** 10.3390/jcm11061641

**Published:** 2022-03-16

**Authors:** Maria Rinzivillo, Francesco Panzuto, Gianluca Esposito, Edith Lahner, Alberto Signore, Bruno Annibale

**Affiliations:** 1Digestive Disease Unit, ENETS Center of Excellence, Sant’Andrea University Hospital, Via di Grottarossa 1035, 00189 Rome, Italy; mariarinzivillo@gmail.com (M.R.); fpanzuto@ospedalesantandrea.it (F.P.); gianluca.esposito@uniroma1.it (G.E.); edith.lahner@uniroma1.it (E.L.); 2Department of Medical-Surgical Sciences and Translational Medicine, “Sapienza” University of Rome, Via di Grottarossa 1035, 00189 Rome, Italy; alberto.signore@uniroma1.it; 3Nuclear Medicine Unit, ENETS Center of Excellence, Sant’Andrea University Hospital, Via di Grottarossa 1035, 00189 Rome, Italy

**Keywords:** gastric carcinoids, neuroendocrine tumors, management, chronic atrophic gastritis

## Abstract

Background: Type I gastric neuroendocrine neoplasia (gNEN) is a rare and low-grade tumor in which the therapeutic strategy is almost always endoscopic. For this reason, the use of radiology or nuclear medicine imaging is not recommended by guidelines. Conversely, in a small number of cases, locoregional or distant metastases may develop, thus suggesting a role for imaging techniques. This retrospective study was performed to explore the usefulness of [^68^Ga]Ga-DOTA-SST PET/CT in the management of patients with T1gNENs. Patients and Method: Single-center retrospective analysis, in an ENETS Center of Excellence, of patients with type I gNEN who underwent [^68^Ga]Ga-DOTA-SST PET/CT. The indication for performing [^68^Ga]Ga-DOTA-SST PET/CT was generally based on the presence of at least one of the following criteria: (1) polyps > 10 mm; (2) endoscopic positive (R1) margin after previous endoscopic resection; and (3) Ki-67 > 3%. Results: A total of 120 patients with T1gNEN were evaluated. Overall, 15 out of 120 (13%) patients had performed [^68^Ga]Ga-DOTA-SST PET/CT. The median Ki-67 value was 6% (IQR 1–9): 9 out of 15 (60%) were G1 tumors, and 6 out of 15 (40%) were G2 tumors. Ninety-three percent of patients were treated by tumor endoscopic resection, whereas surgery was performed in two patients (13%) after incomplete endoscopic resection; the remaining patients (6.6%) received somatostatin analogs due to the presence of multiple recurrent tumors. Overall, [^68^Ga]Ga-DOTA-SST PET/CT was positive in 8 out of 15 patients (53%). Following the [^68^Ga]Ga-DOTA-SST PET/CT findings, the clinical management was modified in 6 out of 15 (40%) patients. Conclusion: [^68^Ga]Ga-DOTA-SST PET/CT can be useful in a restricted and selected group of patients with gastric neuroendocrine neoplasia with relevant risk factors to establish the most appropriate therapeutic strategy.

## 1. Background

Gastric neuroendocrine neoplasms (g-NENs) are rare neoplasms accounting for less than 3% of all gastrointestinal NENs. They are classified on a clinical basis according to the co-existence of a gastric pathology; specifically, when chronic atrophic gastritis (CAG) is diagnosed, they are called “type I gNENs” (T1gNENs) [1].

Although they may occasionally present as metastases [2], their characterization is generally low-grade, with excellent long-term survival and negligible disease-related risk of death.

In fact, the proportion of patients with T1gNEN who develop metastases is <5%, and the long-term overall survival rate is almost 100% [3,4,5]. Thus, endoscopic resection is usually considered a curative treatment for these tumors and rarely requires surgery, which needs to be considered when locoregional invasion of the gastric wall or metastatic lymph nodes is present [6,7,8]. In smaller lesions < 10 mm, observational endoscopic management may also be advised [1,9].

Beyond tumor type, international guidelines recommend evaluating additional factors when facing gNENs [1,10,11]. Among these, the most relevant are tumor size and Ki67 proliferative index, which determines tumor grading (G1: Ki67 < 3%; G2: Ki67 = 3–20%: G3: Ki67 > 20%) [12].

Owing to the higher diagnostic accuracy (sensitivity 91–95%; specificity 82–87%) compared to standard radiological procedures, the hybrid positron emission tomography with ^68^Ga-labeled SST analogs/computed tomography ([^68^Ga]Ga-DOTA-SST PET/CT) is considered the gold standard imaging procedure in GEP-NENs [13]. This technique is widely used for detecting and staging NENs, owing to the expression of somatostatin receptors in the majority of these tumors.

However, given the low grade and the minimal risk of metastatic spread of T1gNENs, their use in this particular setting of NENs is not recommended unless large tumors not suitable for endoscopic resection are found.

Given the rarity of this clinical scenario, data on [^68^Ga]Ga-DOTA-SST PET/CT in type I gastric NENs are lacking. This case series was collected with the intention of exploring the possible utility of [^68^Ga]Ga-DOTA-SST PET/CT in the management of patients with T1gNENs.

## 2. Patients and Method

This is a single-center retrospective analysis including patients evaluated at the Rome (European Neuroendocrine Tumors Society) ENETS Center of Excellence—Sant’Andrea University Hospital, with histologically proven type I gNENs in whom [^68^Ga]Ga-DOTA-SST PET/CT was performed from January 2009 to May 2021. After checking the NEN patients database of the center, all patients with a histologically proven diagnosis ofT1gNEN were included in the study. Among these, those in whom [^68^Ga]Ga-DOTA-SST PET/CT has been performed were selected and included in the final analysis. Additional inclusion criteria were: available data on tumor size and Ki67 value, and data on patients’ follow-up performed at the center according to a specific surveillance program performed in patients with CAG or previous diagnosis of T1gNENs [14,15] (Figure 1).

Gastric NEN was classified as “type I” when CAG was confirmed based on histological examination of atrophy on multiple biopsies performed in the gastric antrum and body. Patients with gNEN other than type 1 were excluded from the study.

Tumors were graded according to the ENETS grading system as follows: G1 (Ki67 < 3%), G2 (Ki67 = 3–20%), and G3 (KI67 > 20%) [12].

Although criteria for selecting patients in whom performing [^68^Ga]Ga-DOTA-SST PET/CT was not standardized, it was required when in the presence of at least one of the following criteria: (1) polyps > 10 mm; (2) positive endoscopic margin after previous endoscopic resection (R1); and (3) Ki-67 > 3%.

The study was conducted according with the Declaration of Helsinki, and full informed consent for data collection was obtained from all the included patients. Data are expressed as median value and interquartile range (25th–75th IQR).

## 3. Results

Among a total of 120 patients with T1gNEN evaluated, [^68^Ga]Ga-DOTA-SST PET/CT was performed in 15 (13%) patients, which were included in the final analysis. Of these, 11 (74%) had been referred from other hospitals where initial gNEN diagnosis has previously been done, whereas the remaining four patients (26%) were diagnosed with developing gNEN during endoscopic surveillance performed for CAG.

The key features of the study population are summarized in Table 1.

Median Ki-67 was 6% (IQR 1–9). Specifically, 9 out of 15 (60%) were G1 tumors, and 6 out of 15 (40%) were G2 tumors. No G3 tumor was found.

Most patients were managed by tumor endoscopic resection, which was planned as the initial treatment in 14 patients (93%). Of these, two patients (13%) also received gastric surgery after incomplete endoscopic resection, whereas the remaining patient (6.6%), who was unfit for surgical resection, received somatostatin analogs due to the presence of multiple recurrent tumors.

Indications to perform [^68^Ga]Ga-DOTA-SST PET/CT were tumor size > 10 mm in 10 patients (66%), uncompleted initial endoscopic resection (R1 margins) in 5 patients (33%), and the presence of Ki67 > 3% (G2 NENs) in 6 patients (40%).

Overall, [^68^Ga]Ga-DOTA-SST PET/CT was positive in 8 out of 15 patients (53%) (Table 2). Specifically, beyond the positivity related to the identification of the primary tumor (*n* = 5, 33%), [^68^Ga]Ga-DOTA-SST PET/CT revealed locoregional or distant metastases in three patients (20%) (liver lesions in one patient, lymph node lesions in two patients) (Figure 2).

Following the [^68^Ga]Ga-DOTA-SST PET/CT findings, the clinical management was modified in 6 out of 15 (40%) patients (Figure 3). Specifically, one patient with liver metastases was treated with somatostatin analogs, and two patients with lymph node metastases were surgically resected with curative intent; in the remaining three patients, in whom total gastrectomy with lymph node removal was initially proposed, the surgical plan was modified to a less aggressive gastric wedge resection with lymph node sampling (one patient) or endoscopic management by endoscopic submucosal dissection (two patients) after [^68^Ga]Ga-DOTA-SST PET/CT had excluded the presence of extragastric tumor lesions, as confirmed by following cross-sectional abdominal imaging.

## 4. Discussion

This study suggests that, in a subgroup of patients with T1gNEN, [^68^Ga]Ga-DOTA-SST PET/CT may help physicians carefully plan therapeutic strategies and disease management.

It is recognized that the vast majority of T1gNENs present as low-grade and do not require aggressive treatments (i.e., surgical procedures and systemic medical anti-proliferative treatments), thus making endoscopic management safe and effective in these patients. However, since distant or locoregional metastases have rarely been reported in a small proportion of patients [2], early identification of these cases is needed to manage them similarly to what is currently done in other digestive NENs.

Although observed in a retrospective single-center study, these results reveal a potential advantage of using [^68^Ga]Ga-DOTA-SST PET/CT in patients with a tumor size > 1 cm, in those cases when histological margins are not tumor-free after endoscopic resection (R1), or when Ki67 is >3% (G2 tumors).

Tumor size is the main prognostic factor, which may be considered the key feature driving physicians to properly manage therapeutic approach and surveillance strategy in patients with T1gNEN [1,3].

In fact, a size > 10 mm has been proposed as the cutoff diameter level to identify tumors that need radical resection, whereas active endoscopic surveillance without resection has been advised for smaller tumors [9].

Conversely, the meaning of Ki-67 in the clinical setting of T1gNENs is still debated [16,17].

Compared to pancreatic and intestinal NENs [18,19], Ki67 has a less clear impact in T1gNEN management and may be considered a minor prognostic factor, a figure that is similar to what is already reported in appendiceal NENs [20].

In fact, in recent studies, Ki67 was unable to discriminate between G1 (Ki67 ≤ 2%) and G2 (Ki67 3–20%) tumors in terms of long-term survival probability [3,4]. Furthermore, in a series of metastatic T1gNEN [2], it varied from 1% to 20%, thus highlighting the lack of a direct relationship between its value and risk of disease aggressiveness.

Regarding radical endoscopic removal, it is clear that achieving margin-free resection is advisable in all patients; however, it is not always achieved irrespective of the endoscopic technique used for tumor removal [8]. Although it is not clear whether residual microscopic tumors may have a direct impact on the disease course, it is reasonable to consider those patients in whom complete resection has not been achieved at higher risk for tumor progression.

The possibility of developing lymph node metastases in patients with T1gNENs is only occasionally observed; however, it is reasonable to consider that the abovementioned risk factors could be relevant, thus suggesting disease staging with [^68^Ga]Ga-DOTA-SST PET/CT in T1gNEN with risk progression features. The ability of this imaging technique to modify NET patient management is well-known. In a previous report, [^68^Ga]Ga-DOTA-SST PET/CT resulted in management modification in up to 45% of patients, given its accuracy to identify distant or locoregional tumor lesions [21]. Although the present study was affected by the limited number of included patients, the retrospective study design (features that are commonly observed in most studies evaluating homogeneous populations of NETs), and the lack of standardized indication for performing [^68^Ga]Ga-DOTA-SST PET/CT, suggests a potential benefit of using this diagnostic tool in selected cases of T1gNENs.

Even though T1gNENs are generally considered low-grade tumors with a favorable clinical course and are considered, in most cases, to not require surgery or systemic medical therapies, attention should be given in selected cases of type I gNENs (size > 1 cm, positive margins after endoscopic resection, Ki67 > 3%), owing to the potential risk of a more aggressive disease. In these patients, an accurate evaluation in the setting of an NEN-dedicated multidisciplinary team is mandatory to decide whether performing [^68^Ga]Ga-DOTA-SST PET/CT, to exclude the presence of a more advanced disease.

However, a prospective study designed to evaluate the diagnostic accuracy of [^68^Ga]Ga-DOTA-SST PET/CT in T1gNEN with an elevated risk of progression is required to better understand the real impact of this technique in the management of these patients.

## Figures and Tables

**Figure 1 jcm-11-01641-f001:**
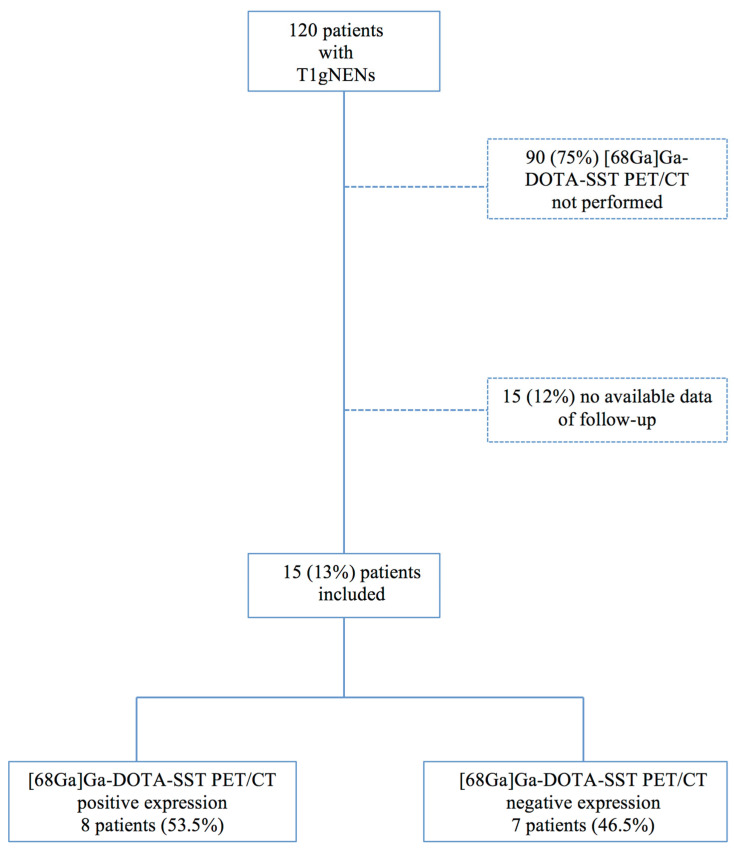
Flow chart of patient selection.

**Figure 2 jcm-11-01641-f002:**
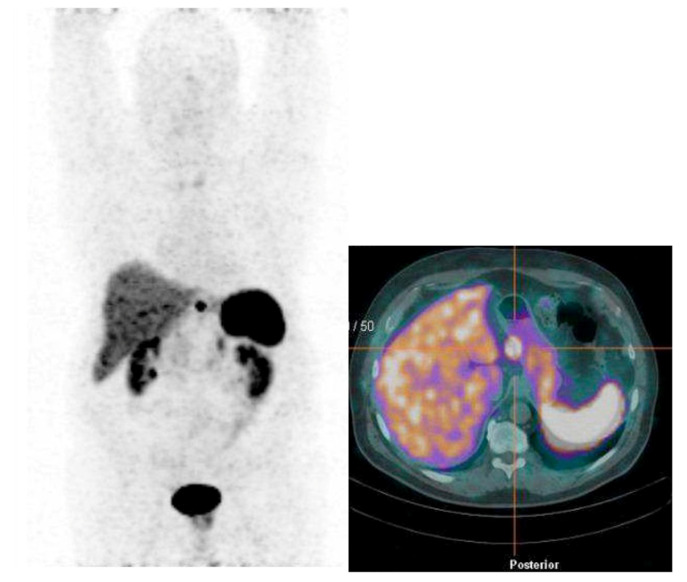
Lymphnodes metastasis in Type I Gastric Neuroendocrine neoplasia evidenced by [^68^Ga]Ga-DOTA-SST PET/CT.

**Figure 3 jcm-11-01641-f003:**
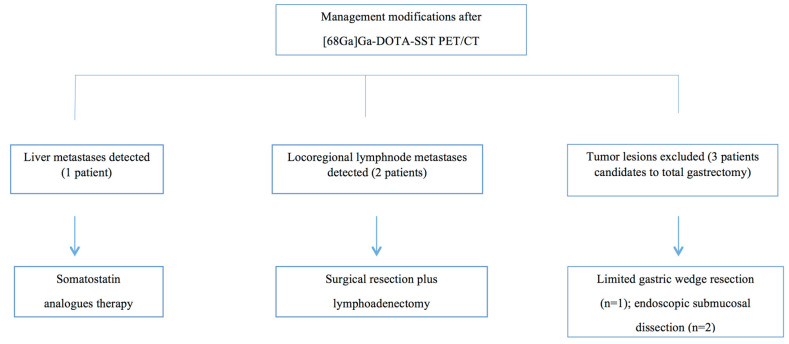
Clinical management changes.

**Table 1 jcm-11-01641-t001:** Patient Characteristics.

	*n*
Gender	8 Female (53%) 7 Male (47%)
Median Age (years)	51 (IQR 49–68)
Ki-67 Proliferative Index	6 % (1–9%)
Grading * G1 G2	9 (60%) 6 (40%)
Tumor size (mm)	13 (IQR 5–55)
Multiple tumor at diagnosis	2 (13.3%)

* according to WHO classification. IQR: interquartile range.

**Table 2 jcm-11-01641-t002:** Patient characteristic with [^68^Ga]Ga-DOTA-SST PET/CT positive expression.

	Dimension of Primitive Tumour	Ki-67 Proliferative Index	[^68^Ga]Ga-DOTA-SST PET/CT Positive Expression on Primitive Tumour	[^68^Ga]Ga-DOTA-SST PET/CT Positive Expression on Locoregional Metastasis	[^68^Ga]Ga-DOTA-SST PET/CT Positive Expression on Distant Metastasis	Endoscopical Resection	Positive Margins after Endoscopical Resection	Surgical Resection	Somatostatin Analogue Therapy
Patient 1	8 mm	1%	Yes	No	No	Yes	Yes	No	No
Patient 2	9 mm	2%	Yes	No	No	Yes	Yes	No	No
Patient 3	12 mm	<1%	Yes	No	No	Yes	No	No	No
Patient 4	12 mm	2%	Yes	No	No	Yes	No	No	No
Patient 5	20 mm	2%	Yes	No	No	Yes	No	No	No
Patient 6	50 mm	4%	No	No	Yes	No	-	No	Yes
Patient 7	20 mm	3%	No	Yes	No	No	Yes	Yes	No
Patient 8	20 mm	2%	No	Yes	No	No	Yes	Yes	No

## Data Availability

Data are available upon request.

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
