# Peer review of "Usefulness of 68-Gallium PET in Type I Gastric Neuroendocrine Neoplasia: A Case Series"

_jcm, 2022, doi:10.3390/jcm11061641_

Round 1

Reviewer 1 Report

The authors present a retrospective analysis in one center in patients with gastric NET type I performing PET-GA68 DOTATOC in order to change the clinical management in those patients

The major concern in the paper is the utility of making PET-Ga to patients with type I gastric NETs. In only 3 patients (2,5%) the clinical management change. Probably the authors will obtain similar finding with other techniques such as TC or RMN. 

PET- GA are Not recommended in guidelines for example NCCN guidelines. Most cases should be followed with endoscopic surveillance.

I suggest authors to increase the number of cases o make prospective study.

Author Response

Reviewer #1

The authors present a retrospective analysis in one center in patients with gastric NET type I performing PET-GA68 DOTATOC in order to change the clinical management in those patients

The major concern in the paper is the utility of making PET-Ga to patients with type I gastric NETs. In only 3 patients (2,5%) the clinical management change. Probably the authors will obtain similar finding with other techniques such as TC or RMN. 

PET- GA are Not recommended in guidelines for example NCCN guidelines. Most cases should be followed with endoscopic surveillance.

I suggest authors to increase the number of cases o make prospective study.

Reply

We agree with the reviewer regarding the potential value of increasing the number of patients and/or performing a prospective study, as mentioned in the last sentence of the manuscript. Unfortunately, both those proposals are difficult to realize, as in many other clinical setting involving rare diseases as gNENs are.

We also agree that most type I gNENs may be safely followed-up by endoscopic surveillance, according with the international guidelines, and that the present study has not the strenght to modify this recommendation. However, we believe that, despite the above-mentoned study limitations, performing [68Ga]Ga-DOTA-SST PET/CT may help physicians to properly stage the disease in those patients who have a high risk for developing a more aggressive disease. A sentence to clarify this issue has been added at the end of the discussion (page 4/9)

As far as the opportunity to perform TC or MRI, we believe that choosing [68Ga]Ga-DOTA-SST PET/CT is advisable thanks to its higher diagnostic accuracy, as now mentioned in the introduction (page 2/9)

Reviewer 2 Report

The manuscript by Rinzivillo et al. explored the potential role of gallium-68 PET in the management of patients with type I gastric neuroendocrine neoplasia (gNEN), and found that gallium-68 PET could be useful in a selected and high-risk group of patients with gNENs.

And there are several minor deficiencies that need to be addressed.

  1. The abbreviation “ENETS” should spell out the full term when used for the first time.
  2. In table 1, the row “Dimension of gNENs” should add “mm”, and the abbreviation “IQR” should spell out in full term in the notes below the table.
  3. In the discussion section, the authors stated that “In a previous report, gallium-68PET resulted in a change in clinical management in up to 45% of NEN patients, due to its ability to detect distant or locoregional metastases [22]. Although gallium-68PET was affect-ed by the small number of included patients and the retrospective study design (features that are commonly observed in most studies evaluating homogeneous populations if NETs), the present study suggests a potential benefit of using this diagnostic tool in selected cases of T1gNENs”. There should be a space between “gallium-68” and “PET”.

Author Response

Reviewer #2

The manuscript by Rinzivillo et al. explored the potential role of gallium-68 PET in the management of patients with type I gastric neuroendocrine neoplasia (gNEN), and found that gallium-68 PET could be useful in a selected and high-risk group of patients with gNENs.

And there are several minor deficiencies that need to be addressed.

  1. The abbreviation “ENETS” should spell out the full term when used for the first time.

Reply: thank you for ths comment, the abbreviation ENETS has been explained in the text (page 2/9)

  1. In table 1, the row “Dimension of gNENs” should add “mm”, and the abbreviation “IQR” should spell out in full term in the notes below the table.

Reply: Table has been modified according with the Reviewer’s suggestions. In addition, the raws regarding treatments have been deleted, since that data is already reported in the text

  1. In the discussion section, the authors stated that “In a previous report, gallium-68PET resulted in a change in clinical management in up to 45% of NEN patients, due to its ability to detect distant or locoregional metastases [22]. Although gallium-68PET was affect-ed by the small number of included patients and the retrospective study design (features that are commonly observed in most studies evaluating homogeneous populations if NETs), the present study suggests a potential benefit of using this diagnostic tool in selected cases of T1gNENs”. There should be a space between “gallium-68” and “PET”.

Reply: thank you for this comment, the sentence hase been rephrased

Reviewer 3 Report

Undoubtedly, the work presented for review raises a very interesting topic - cancer therapy. We all know that the search for new, effective and, at the same time, safe methods for the patient is extremely important. 

Despite the very interesting subject matter, the reviewer believes the work has some shortcomings. The chapter "Patients and Methods" was described very laconically. Among other things, it lacks: - flow chart of patient selection for the study - description of the recruitment / selection of patients for the study - operational protocol - which method was used for the statistical evaluation - whether patients' consent to participate in the study was obtained. During the discussion of the results, it should be more emphasized what are the limitations of the proposed method. In summary, the work is interesting, but requires some corrections to be better understood by the reader. 

Author Response

Reviewer #3

Undoubtedly, the work presented for review raises a very interesting topic - cancer therapy. We all know that the search for new, effective and, at the same time, safe methods for the patient is extremely important. 

Despite the very interesting subject matter, the reviewer believes the work has some shortcomings. The chapter "Patients and Methods" was described very laconically. Among other things, it lacks: - flow chart of patient selection for the study - description of the recruitment / selection of patients for the study - operational protocol - which method was used for the statistical evaluation - whether patients' consent to participate in the study was obtained. During the discussion of the results, it should be more emphasized what are the limitations of the proposed method. In summary, the work is interesting, but requires some corrections to be better understood by the reader. 

Reply: we thank the Reviewer for the valuable comments, which were used for improving the section “patients and methods” of the manuscript (page 2/9)

In addition, the study limitations have been emphasized in the discussion, as suggested (page 4/9)

Figure 1 reporting the patients’ flow chart has been added.

Round 2

Reviewer 1 Report

The authors try to improve the paper. The same concerns about still on air. I recommend to include more patients. The paper is not really relevant to clinical management in those patient. 

Reviewer 3 Report

The authors referred to the reviewer's doubts. The work has been improved sufficiently.